# Thyroid Autoimmunity in CSU: A Potential Marker of Omalizumab Response?

**DOI:** 10.3390/ijms24087491

**Published:** 2023-04-19

**Authors:** Riccardo Asero, Silvia Mariel Ferrucci, Paolo Calzari, Dario Consonni, Massimo Cugno

**Affiliations:** 1Clinica San Carlo, Ambulatorio di Allergologia, 20037 Paderno Dugnano, 20037 Milan, Italy; 2Unit of Dermatology, Fondazione IRCCS Ca’ Granda Ospedale Maggiore Policlinico, 20122 Milan, Italy; 3Postgraduate School of Allergy and Clinical Immunology, Department of Pathophysiology and Transplantation, Università degli Studi di Milano, 20122 Milan, Italy; 4Epidemiology Unit, Fondazione IRCCS Ca’ Granda Ospedale Maggiore Policlinico, 20122 Milan, Italy; 5Department of Pathophysiology and Transplantation, Università degli Studi di Milano, 20122 Milan, Italy; 6Dipartmento di Medicina Interna, UOC Medicina Generale—Emostasi e Trombosi, Fondazione IRCCS Ca’ Granda Ospedale Maggiore Policlinico, 20122 Milan, Italy

**Keywords:** urticaria, thyroid autoimmunity, atopic status, total IgE, omalizumab

## Abstract

The response of severe chronic spontaneous urticaria (CSU) to omalizumab largely depends on the autoimmune or autoallergic endotype of the disease. Whether thyroid autoimmunity may predict omalizumab response along with total IgE in CSU is still unclear. Three hundred and eighty-five patients (M/F 123/262; mean age 49.5 years; range 12–87 years) with severe CSU were studied. Total IgE levels and thyroid autoimmunity (levels of anti-thyroid peroxidase [TPO] IgG) were measured before omalizumab treatment. Based on the clinical response, patients were divided into early (ER), late (LR), partial (PR) and non (NR) responders to omalizumab. Thyroid autoimmunity was detected in 92/385 (24%) patients. Altogether, 52%, 22%, 16% and 10% of patients were ER, LR, PR and NR to omalizumab, respectively. Response to omalizumab was not associated with thyroid autoimmunity (*p* = 0.77). Conversely, we found a strongly positive association between IgE levels and omalizumab response (*p* < 0.0001); this association was largely driven by early response (OR = 5.46; 95% CI: 2.23–13.3). Moreover, the predicted probabilities of early response strongly increased with increasing IgE levels. Thyroid autoimmunity alone cannot be used as a clinical predictor of omalizumab response. Total IgE levels remain the only and most reliable prognostic marker for omalizumab response in patients with severe CSU.

## 1. Introduction

The severity of chronic spontaneous urticaria (CSU), a rather frequent disease characterized by recurrent wheals with or without angioedema for more than 6 weeks, may show much variability. Most patients show a slight intermittent or recurrent disease that is easily controlled by antihistamines only, whereas a minority of patients are affected by a continuous and severe disease that is completely refractory to antihistamine treatment and that worsens their quality of life dramatically. The introduction, less than one decade ago, of omalizumab, an anti-IgE monoclonal antibody, in CSU therapy has completely re-defined the treatment of severe CSU [1], but has also led to the identification of different endotypes of this disease. Most treated patients show a rapid and complete response to the drug (and for this reason have been called “early responders” (ER)), while a minority of patients with severe CSU respond only after several months (the so-called “late responders” (LR)) or seem completely refractory to treatment (“non responders” (NR)). This variable behavior has been associated with the existence of different underlying patho-mechanisms of the disease, the former patients probably having an IgE-mediated autoimmune disease (defined as “autoallergic” or type I CSU) and the latter patients a typical IgG-mediated autoimmune disease (defined as type IIb CSU). In type I CSU, mast-cell-bound autoimmune IgE is able to react to a large spectrum of autoallergens [2], whereas in type IIb CSU, IgG specific for IgE or for the high-affinity IgE receptor is involved [3,4]. Both mechanisms are eventually able to lead to mast cell degranulation and histamine release. In view of the different responses to anti-IgE therapy, it is not surprising that a number of studies have sought pathogenic markers that may be practically helpful to predict patients’ response to omalizumab. For instance, positive in vitro tests of basophil activation by sera from CSU patients (either the basophil activation test (BAT) or the basophil histamine release assay (BHRA)), or the presence of antinuclear antibodies have been associated with a late response/non-response to omalizumab and are therefore considered as markers of autoimmune urticaria [5,6,7,8]. Similarly, the association of a positive autologous serum skin test (ASST) and of low levels of circulating IgE predicts a late or poor response to anti-IgE treatment [7,9]. Conversely, there is large consensus that elevated levels of total IgE are frequently associated with a prompt response to omalizumab [9,10,11,12,13,14,15,16]. The association between CSU and thyroid autoimmunity has been known for almost 30 years [17], and the frequent co-occurrence of different autoimmune disorders in CSU patients is now established [18,19,20]. Although it has been observed that a typical type IIb CSU patient is frequently characterized by low IgE and autoimmune thyroiditis [21], whether thyroid autoimmunity alone may predict a poor or absent response to omalizumab in clinical practice has not been clearly defined so far. In a recent study, thyroid autoimmunity was equally prevalent in severe CSU patients with different levels of total IgE [22]. In the present study, we investigated whether the presence of thyroid autoimmunity alone and the level of thyroid peroxidase IgG autoantibodies may be helpful to predict the response to omalizumab in clinical practice, thus discriminating patients with probable severe type I or type IIb CSU.

## 2. Results

Of 385 patients, 92 (24%) scored positive for IgG to TPO. On omalizumab treatment, 200 patients (52%) were defined as ER, 84 (22%) as LR, 39 (10%) as NR, and 62 (16%) as PR. In total, 92/385 (24%) were diagnosed as showing thyroid autoimmunity. Response to omalizumab was not associated with gender (*p* = 0.38), age category (*p* = 0.22) or thyroid autoimmunity (Table 1 and Figure 1).

Among the 92 anti-TPO-positive patients, precise levels of IgG to TPO were available for 44 patients. No statistical differences in omalizumab response were detected between patients showing low (60–500 IU/mL; n = 22) and high (>500 IU/mL; n = 22) levels of anti-thyroid peroxidase IgG (Table 2).

Baseline total IgE levels were available for 316 patients. These exceeded 100 IU/mL in 146 (46%) cases. We found a positive association of omalizumab response with dichotomized (low/high) IgE levels (Table 3 and Figure 2), mainly driven by a strong association with early response (OR = 5.46). The predicted probabilities of early response strongly increased with increasing IgE levels (Figure 3). These associations were confirmed in a multivariable (adjusted for gender and age category) multinomial model: partial response OR 4.45 (95%CI: 1.92–10.5), late response OR 3.10 (95% CI: 1.50–6.43) and early response OR 6.25 (95% CI: 3.11–12.5).

## 3. Discussion

CSU has been found to be associated with autoimmune diseases [18,19,20], and several lines of evidence support the fact that chronic urticaria shows a preferential association with autoimmune thyroid disorders [17,23,24], although the reasons for this are still unclear [24]. In effect, the prevalence of thyroid autoimmunity in our study group (24%) was even higher than the one found in CSU patients in previous studies [17,23,24]. This difference might be due to the fact that our cohort was represented by CSU with a particularly severe disease refractory to antihistamine treatment. Thyroid autoimmunity has been considered indicative of a type IIb autoimmune CSU phenotype [25,26], and the recent PURIST study found that type IIb CSU patients tend to have high levels of TPO autoantibodies [4]. In a recent study, Kolkhir and co-workers [26] found that about 10% of CSU patients show the presence of elevated TPO IgG autoantibodies and low levels of total IgE. These subjects are preferentially females and show a severe disease of short duration that appears at an older age, along with a high prevalence of positive autologous serum skin test, positive basophil activation test (BAT) scores and a poorer response to antihistamines. Our study cannot be directly compared with that of Kolkhir et al. [26], as we examined a selected population of CSU patients with a severe disease not responding to antihistamine treatment. Nonetheless, the results of our study strongly suggest that thyroid autoimmunity on its own cannot be considered as a clinically useful marker of type IIb CSU or a predictive marker of poor response to omalizumab, as its prevalence was the same in all CSU subgroups. These findings confirm and enforce the observations of a previous recent study carried out in one of our centers [22].

Altogether, this study, albeit limited to the investigation of one single parameter in a specific subset of chronic urticaria patients, confirms that the detection of type IIb CSU patients requires the concordance of a series of different tests, including the autologous serum skin test, in vitro basophil activation tests and positive IgG anti-FcεRI or anti-IgE. Thus, type IIb CSU patients are not easily detectable in normal clinical settings [7]. In conclusion, thyroid autoimmunity alone does not seem of particular help in the stratification of CSU patients, and baseline total IgE levels currently remain the only and most easily available prognostic marker for omalizumab response in patients with severe CSU.

## 4. Materials and Methods

### 4.1. Patients

Three hundred and eighty-five patients (M/F 123/262; mean age 49.5 years; range 12–87 years) with severe CSU unresponsive to antihistamines at any dosage and prescribed omalizumab were studied. All patients were visited and followed up at the Clinica San Carlo and the Department of Dermatology of the University of Milan. The mean disease duration was 58 months (range 2–720 months), and baseline UAS-7 level was >28 in all cases.

Following the current Italian legislation, patients were administered a fixed dose of 300 mg of omalizumab given subcutaneously at 4-week intervals, and those unresponsive after 3 months had to stop the treatment. Non-response (NR) to omalizumab was defined as the absence of any change (i.e., a reduction of at least 50%) in UAS-7 values 3 months after the start of the treatment. A late response (LR) was diagnosed if a reduction > 50% of UAS-7 score occurred within 1 to 3 months after the first administration of the drug. An early response (ER) to omalizumab was defined as the disappearance or a reduction > 50% in the UAS-7 score within 4 weeks after the first administration. Patients in which a reduction > 50% in UAS-7 occurred under omalizumab but no further improvement was recorded were considered partial responders (PR).

### 4.2. Methods

Anti-TPO IgG autoantibodies and total IgE were measured by commercial enzyme fluoroimmune assays, EliA anti-TPO and EliA total IgE, Thermo Fisher Scientific (Waltham, MA, USA).

Patients were considered as having thyroid autoimmunity in the presence of a positive anti-TPO assay, irrespective of their thyroid functional status. Values > 60 IU/mL were considered positive.

The internal review board of the participating centers approved this retrospective, anonymous study.

### 4.3. Statistical Analysis

To analyze the response to omalizumab according to the levels of thyroid peroxidase IgG autoantibodies (high vs. normal) and IgE (high vs. normal), we used chi-squared tests and fitted multinomial (polytomous) logistic regression models to calculate the odds ratios (OR) and confidence intervals (CI) of early, late or partial response, considering the absence of response as a reference. In addition, we fitted a multinomial logistic model to analyze the association of total IgE, considered as a continuous variable, to obtain the predicted probabilities of response. Finally, we fitted a multivariable (adjusted for gender and age category) multinomial model to total IgE, again considered as a continuous variable. Statistical analysis was performed with Stata 17 (StataCorp. 2021, College Station, TX, USA).

### 4.4. Ethics

Patients gave informed written consent to the use of their clinical data in anonymous form. The internal review board of the Clinica San Carlo approved the study in April 2022. Since the study was exclusively observational and based only on routine analyses, a formal approval by an external ethical committee was not warranted or requested.

## Figures and Tables

**Figure 1 ijms-24-07491-f001:**
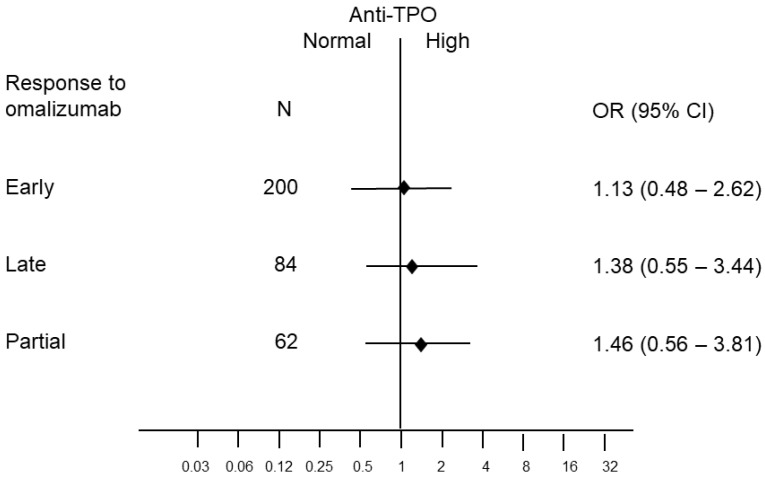
Odds ratios (OR) and 95% confidence intervals of early, late and partial response to omalizumab according to anti-thyroid peroxidase (TPO) IgG autoantibody levels (high vs. low) from a multinomial logistic regression model. Reference: no response.

**Figure 2 ijms-24-07491-f002:**
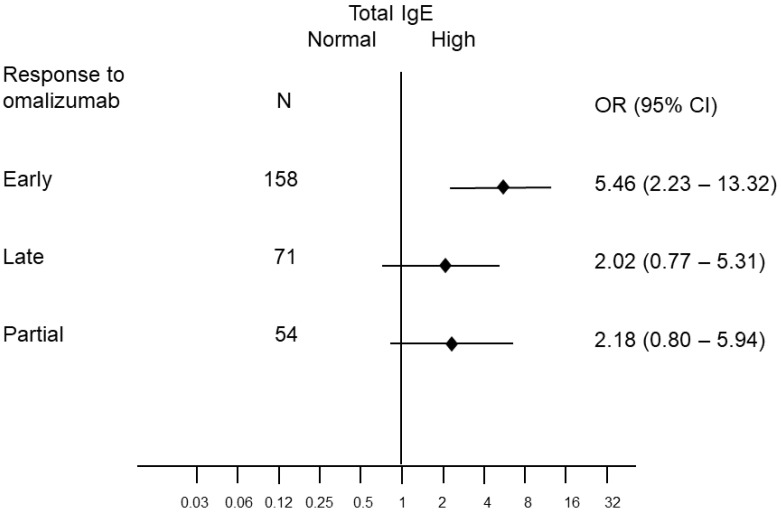
Odds ratios (OR) and 95% confidence intervals of partial, late and early response to omalizumab according to IgE levels (high vs. low) from a multinomial logistic regression model. Reference: no response.

**Figure 3 ijms-24-07491-f003:**
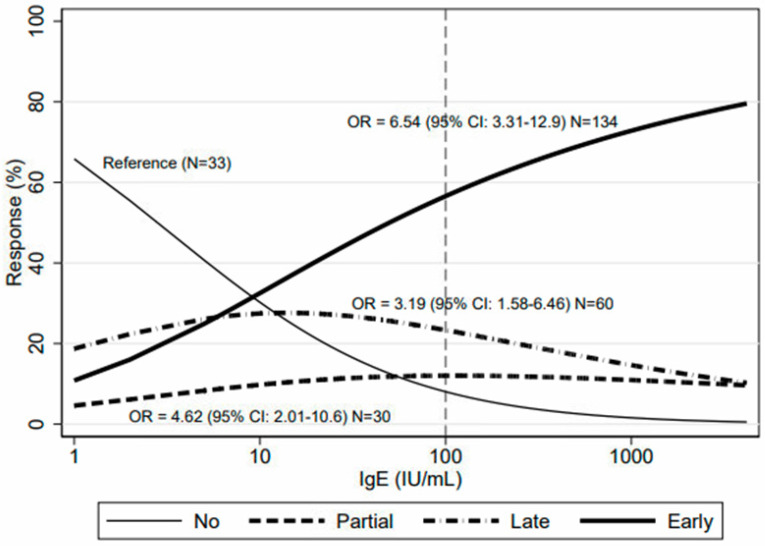
Predicted probability (%) of no, partial, late or early response to omalizumab according to IgE levels (continuous) from a multinomial logistic regression.

**Table 1 ijms-24-07491-t001:** Clinical response to omalizumab in 385 urticaria patients with and without thyroid autoimmunity.

Study Population (n = 385)	Thyroid + (n = 92)	Thyroid − (n = 293)
Non-response (n = 39)	8 (9%)	31 (11%)
Partial response (n = 62)	17 (18%)	45 (15%)
Late response (n = 84)	22 (24%)	62 (21%)
Early response (n = 200)	45 (49%)	155 (53%)

*p*-value = 0.77 (chi-squared test).

**Table 2 ijms-24-07491-t002:** Clinical response to omalizumab as a function of the level of thyroid peroxidase IgG.

Patients (n = 44)	Weak Thyroid Autoimmunity (60–500 IU/mL; n = 22)	Strong Thyroid Autoimmunity (>500 IU/mL; n = 22)	*p*
Early response	14	8	NS
Late response	5	6	NS
Non-response	0	3	NS
Partial response	3	5	NS

NS = not statistically significant.

**Table 3 ijms-24-07491-t003:** Clinical response to omalizumab in patients showing different baseline IgE levels.

Study Population (n = 316)	IgE >= 100 (n = 146)	IgE < 100 (n = 170)
Non-response (n = 33)	7 (5%)	26 (15%)
Partial response (n = 54)	20 (14%)	34 (20%)
Late response (n = 71)	25 (17%)	46 (27%)
Early response (n = 158)	94 (64%)	64 (38%)

*p*-value < 0.0001 (chi-squared test).

## Data Availability

Data available in electronic format from the authors.

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
