# Peer review of "Thyroid Autoimmunity in CSU: A Potential Marker of Omalizumab Response?"

_ijms, 2023, doi:10.3390/ijms24087491_

Round 1
Reviewer 1 Report
This clinical (observational) study was addressed to verify whether the response of severe chronic spontaneous urticaria (CSU) to omalizumab (OMZ) was dependent on thyroid autoimmunity and/or total IgE levels. The study was carried out on 385 patients with CSU (unresponsive to antihistamines, characterised by a mean duration of 58 months and defined by a baseline UAS-7 level > 28 in all cases). Omalizumab was administered at a fixed dose of 300 mg at 4-week intervals and patients were classified (with reference to the UAS-7 score) as early (ER)-, partial (PR-), late (LR)- and non-responders (NR). Thyroid autoimmunity was evaluated by determining the level (> 60 IU/ml) of anti-TPO IgG autoantibodies, while also total IgE level was determined by an enzyme fluoroimmune assay. It was concluded that the response to OMZ was not related to gender, age of thyroid autoimmunity, being instead (positively) associated with the IgE levels (whose increase appeared to favour an early OMZ response).
In Introduction, Authors distinguish between Type I CSU (“autoallergic”) and Type IIb CSU (equally mediated by IgG), but this nosographic difference does not result to have been adequately discussed although CSU is likely to involve a series of different pathogenetic determinants. Moreover, Authors evidence how there are conflicting literature data linking, in CSU patients, plasma levels of IgE auto-antibodies (mostly against various thyroid constituents including nuclei) and/or total IgE levels to the response to OMZI.
In Results, it is pointed out that low or high levels of anti-TPO IgG did not influence OMZ response, whereas baseline increasing total IgE levels favoured an early OMZ response.
In Discussion, Authors underline how type IIb autoimmune CSU phenotype cannot be defined by the only presence of thyroid autoimmunity which, by itself, does not allow to predict a poor OMZ response. Indeed, a series of different concordant tests are required to give credit to the diagnosis of type IIb CSU.
Some remarks:
- Authors should indicate the route of administration (subcutaneous?) of OMZ (300 mg).
- Authors should better clarify the differences between the type I CSU (“autoallergic”) and the type IIb CSU since, in Introduction, both types were defined as “…IgG-mediated autoimmune disease… typical or not typical…?...
- By which (autoimmune) mechanisms IgE are able to provoke the lesions that characterise CSU (Introduction: wheals=weals!)? The presence of weals, angioedema etc. might suggest the involvement of complex immune mechanisms not only attributable to a simple IgE increase. Some explanations on these points would be useful to improve the manuscript, thus pinpointing the CSU effective or plausible pathogenesis and stimulating the readers’ interest.
- The manuscript should be thoroughly checked since, here and there, there are several lexical imprecisions not exactly corresponding to an adequate “English style”. References and figures are acceptable.
Author Response
Responses to Reviewer #1
We thank the reviewer for the useful remarks that have allowed us to improve the manuscript.
COMMENT: Authors should indicate the route of administration (subcutaneous?) of OMZ (300 mg).
RESPONSE: In the methods section we have pointed out that the drug was administered subcutaneously
COMMENT: Authors should better clarify the differences between the type I CSU (“autoallergic”) and the type IIb CSU since, in Introduction, both types were defined as “…IgG-mediated autoimmune disease… typical or not typical…?...
RESPONSE: In the introduction of the original submission we wrote “This variable behavior has been associated with the existence of different underlying patho-mechanisms of the disease, the former probably having an IgE-mediated autoimmune disease (defined as “autoallergic” or Type I CSU) and the latter a typical IgG-mediated autoimmune disease (also defined Type IIb CSU). Both mechanisms are eventually able to lead to mast cell degranulation and histamine release”. It appears quite clear to us that albeit both mechanisms are autoimmune in origin, the former (autoallergic) is IgE-driven, while the second (type IIb) is IgG-mediated.
COMMENT: By which (autoimmune) mechanisms IgE are able to provoke the lesions that characterise CSU (Introduction: wheals=weals!)? The presence of weals, angioedema etc. might suggest the involvement of complex immune mechanisms not only attributable to a simple IgE increase. Some explanations on these points would be useful to improve the manuscript, thus pinpointing the CSU effective or plausible pathogenesis and stimulating the readers’ interest.
RESPONSE: In the revised manuscript we have explained more in detail how IgE and IgG are able to induce the degranulation of mast cells. To this end, 3 new references have been added.
COMMENT: The manuscript should be thoroughly checked since, here and there, there are several lexical imprecisions not exactly corresponding to an adequate “English style”. References and figures are acceptable
RESPONSE: We have thoroughly revised the manuscript for grammar errors and imprecisions.

Reviewer 2 Report
While the article is short and concise, but it was comprehensive and sufficient.
However, the discussion was too short. Suggest adding limitation and conclusion at the end.
Table 1 remove extra column
It is unclear whether the regression models included covariates as age, sex, ....
Author Response
Responses to Reviewer #2
We thank the reviewer for the useful comments that have led to improve the manuscript.
COMMENT: However, the discussion was too short. Suggest adding limitation and conclusion at the end.
RESPONSE: The discussion has been enlarged, with also the addition of some new references. The limitations of the study have been acknowledged at the end of the discussion and the conclusions have been added.
COMMENT: Table 1 remove extra column
RESPONSE: Done, thank you for the remark.
COMMENT: It is unclear whether the regression models included covariates as age, sex, ....
RESPONSE: In the previous versions we only fitted crude models because the influence of gender and age was weak. But we agree it could be useful to provide adjusted results. In this revised version we added (see Methods and Results sections) the result of a multivariable (adjusted for gender and age category) multinomial logistic model to total IgE (continuous).